# Predictors of Preeclampsia in the First Trimester in Normotensive and Chronic Hypertensive Pregnant Women

**DOI:** 10.3390/jcm12020579

**Published:** 2023-01-11

**Authors:** Susana Vázquez, Julio Pascual, Xavier Durán-Jordà, Jose Luís Hernández, Marta Crespo, Anna Oliveras

**Affiliations:** 1Nephrology Department, Hospital Universitari del Mar, 08003 Barcelona, Spain; 2IMIM, Hospital del Mar Medical Research Institute, 08003 Barcelona, Spain; 3Obstetrics and Gynecology Department, Hospital Universitari del Mar, 08003 Barcelona, Spain; 4Department of Experimental and Health Sciences, Area of Medicine, Universitat Pompeu Fabra, 08002 Barcelona, Spain; 5Red de Investigación Renal (REDINREN), Instituto Carlos III-FEDER, 28029 Madrid, Spain

**Keywords:** preeclampsia, predictive factors, office blood pressure, ambulatory blood pressure monitoring, chronic hypertension

## Abstract

Preeclampsia (PE) is characterized by the new onset of hypertension (HT) and proteinuria beyond the 20th week of gestation. We aimed to find the best predictor of PE and find out if it is different in women with or without HT. Consecutively attended pregnant women were recruited in the first trimester of pregnancy and followed-up. Laboratory and office and 24 h-ambulatory blood pressure (BP) data were collected. PE occurred in 6.25% of normotensives (*n* = 124). Both office mean BP and 24 h-systolic BP in the first trimester were higher in women with versus those without PE (*p* ≤ 0.001). In women with chronic hypertension (cHT), PE occurred in 55%; office SBP (*p* = 0.769) and 24 h-SBP (*p* = 0.589) were similar between those with and those without PE. Regarding biochemistry, in cHT, plasma urea and creatinine were higher in PE women than in those without cHT (*p* = 0.001 and *p* = 0.004 for the differences in both parameters). These differences were not observed in normotensives. In normotensives, mean BP was the best predictor of PE [ROC curve = 0.91 (95%CI 0.82–0.99)], best cut-off = 80.3 mmHg. In cHT, plasma urea and creatinine were the best predictors of PE, with ROC curves of 0.94 (95%CI 0.84–1.00) and 0.93 (95%CI 0.83–1.00), respectively. In the first trimester of pregnancy, the strongest predictor of PE in normotensive women is office mean BP, while in cHT, renal parameters are the strongest predictors. Otherwise, office BP is non-inferior to 24 h ambulatory BP to predict PE.

## 1. Introduction

Preeclampsia (PE) is a multi-system progressive disorder characterized by the new onset of hypertension and proteinuria or significant end-organ dysfunction after the 20th week of gestation [1]. The prevalence of PE is around 2–8% of pregnancies worldwide [2] and it is a major cause of maternal and perinatal mortality and morbidity. Chronic hypertension (cHT) is estimated to affect 0.9–1.5% of pregnant women [3]. The main risk for pregnant women with cHT is the development of superimposed preeclampsia (sPE), which occurs in 25% of cases [4]. The rate of maternal cHT has increased in recent years, largely explained by the increase in obesity and maternal age [5]. In recent years, several studies have indicated that a combination of maternal history and biochemical and biophysical markers effectively predicts PE in the first trimester of pregnancy [6,7] and thus allows early initiation of prophylactic treatment with acetylsalicylic acid. This treatment is effective if administered within the first 16 weeks of gestation [8].

HT in PE is a consequence of vasoconstriction phenomena and increased peripheral vascular resistance. So far, the best-known predictor of PE is mean arterial blood pressure (MBP), which increases at early stages of gestation [9]. Furthermore, the value of MBP obtained in the first trimester is directly proportional to the chronology and severity of the disorder [10]. However, in pregnant women with cHT, MBP provides contradictory and poor capacity to predict PE in different studies [11,12]. Overall, the data published until now suggest that in pregnant women with cHT there is no clear predictor for the development of sPE.

In this study, we aimed to assess which hemodynamic or biochemical parameters determined in the first trimester predict PE, and whether they are different in normotensive and cHT pregnant women. In addition, we have looked for possible differences in the predictive value of BP as measured in an office or ambulatory setting.

## 2. Materials and Methods

This prospective cohort study was conducted at the Hypertension and Vascular Risk Unit of the Nephrology Department of the Hospital del Mar in Barcelona from December 2015 to June 2018. Participants were recruited at the time of the routine first-trimester scan (13.7 ± 2 weeks of gestation). Exclusion criteria were multiple pregnancy, age below 18 years and chronic kidney disease. A total of 144 singleton pregnant women were eligible to enter the study and provided the written informed consent. The local ethics committee approved the study protocol

Arterial BP was measured by using an automatic calibrated device (Digital Blood Pressure Monitor Model HEM-907 XL IntelliSense^®^ (Omron, USA, Lake Forest, IL, USA). BP measurement was performed according to the European Society of Hypertension Guidelines for office blood pressure measurement [13]. Participants were sitting resting for 10 min in a quiet room. Three consecutive readings were taken, separated by 2-min intervals and with the cuff appropriately sized to fit the arm circumference. The final clinical BP was obtained from the average of these 3 measurements. Data were collected on systolic blood pressure (SBP) and diastolic blood pressure (DBP). Mean arterial blood pressure was calculated as DBP + (SBP − BP)/3. A 24-h ambulatory BP monitoring (24 h-ABPM) was then performed using a SpaceLabs 90207 device, scheduled to measure BP every 20 min during the daytime and every 30 min during the nighttime. The waking and sleeping periods were established according to each individual report. Twenty-four-hour ABPM recordings were considered successful when the percentage of the measurements was >70%, with at least one valid measurement every hour. Fasting venous blood and urine samples were obtained the same day.

Preeclampsia was diagnosed by a nephrologist expert in hypertension according to the guidelines of the International Society for the Study of Hypertension in Pregnancy: SBP ≥ 140 mmHg and/or ≥90 mmHg, confirmed by repeated measurements over a few hours, developing after 20 weeks of gestation in previously normotensive women, accompanied by proteinuria of urine protein/creatinine ratio ≥ 0.3 mg/mg. Superimposed preeclampsia to cHT was defined as the onset of this disorder in a pregnant woman with cHT [14].

### Statistical Analysis

A database was created by using SPSS (version 19.0, Cary, NC, USA). Some specific analyses were also performed using the STATA program. Variables following normal distribution are summarized as mean ± S.D. and categorical data are presented as frequencies and percentages. Data with non-normal distribution are summarized as median and interquartile range (IQR). For normally distributed quantitative variables, groups were compared using the Student’s *t*-test for samples with two categories. For comparison between groups of non-parametric variables, the Mann–Whitney U or Kruskall–Wallis tests were used. The Chi-square test was used for the comparison of categorical variables. The predictive value of the different parameters for PE was evaluated by the area under the ROC curve and its 95% confidence intervals (95%CI). Likewise, based on the estimation of the ROC curve, indicative cut-offs were established for the prediction of PE for each of the parameters with statistically significant differences and the corresponding sensitivity, specificity, and positive and negative predictive values were calculated.

## 3. Results

We evaluated 144 consecutively recruited pregnant women in their first trimester of gestation. Normotensive pregnant women (*n =* 124) and pregnant women with a history of cHT (*n =* 20) were separately analyzed. Sixteen (80%) women of this latter group were on antihypertensive treatment.

Baseline characteristics of normotensive and cHT pregnant women are compared in Table 1. Women with cHT had a significantly higher body mass index (BMI) and the percentage of primiparous women was lower in this group.

The maternal characteristics are separately shown for both PE and sPE in Table 2, comparing those of women with unaffected pregnancies in either the normotensive or the cHT group, respectively. The incidence of PE was 6.4% (8/124) in the normotensive group and 55% (11/20) in the cHT group.

In terms of ethnicity, there is a high prevalence of non-Caucasian pregnant women, which represents a racially heterogeneous population, as corresponds to the population served in our area. Thus, non-Caucasian women account for 30% of normotensive pregnant women and 55% of pregnant women with cHT.

Normotensive pregnant women who developed PE had a higher baseline BMI than those with an unaffected pregnancy. In addition, 87% of normotensive pregnant women who developed PE reported a family history of HT with a statistically significant higher prevalence than women unaffected by PE. Previous PE was more frequent in the pregnant women who developed PE in both normotensive and cHT pregnant women.

The hemodynamic parameters (office and 24-h ambulatory BP) determined in the first trimester of pregnancy are shown in Table 3 and Table 4, respectively. In normotensive pregnant women who eventually developed PE, office and ambulatory SBP, DBP, and MBP values were higher as compared to the group of women with unaffected pregnancies. On the contrary, these differences in BP between women with unaffected pregnancies and those who developed sPE were not observed among pregnant women with cHT. As regards pulse pressure (either office and 24-ambulatory BP), there was not statistically significant differences in pregnant women who subsequently developed preeclampsia compared to those who had an unaffected pregnancy (Appendix A).

Laboratory parameters are shown in Table 5. In normotensive pregnant women, no differences were found between those who were unaffected and those who developed PE. However, in pregnant women with cHT, plasma urea, creatinine, and urate levels were higher in those who developed sPE compared to those with unaffected pregnancies.

The ability of BP parameters (both office and 24-h ambulatory BP) to predict PE is shown in Table 6. This predictive capacity for the development of PE in both normotensive and cHT pregnant women was calculated with the area under ROC curve for BP parameters. Among all the hemodynamic parameters determined in the first trimester, office MBP is the best predictor for the development of PE in normotensive pregnant women with a sensitivity of 100% and a specificity of 74%, with an area under ROC curve of 0.91 (95%CI 0.82–0.99) (Table 7, Appendix A). In addition, the office MBP cut-off point with the highest predictive value for the development of PE in normotensive pregnant women was 80.3 mmHg (Table 7).

Given that the women with cHT who developed sPE had higher values of plasma urea and creatinine concentrations than those with unaffected pregnancies, the accuracy of these parameters to predict sPE was calculated. As shown in Table 8, plasma urea at baseline was a good predictor for the development of added pre-eclampsia, with a sensitivity of 91%, specificity of 87.5% and area under the ROC curve of 0.94 (95%CI 0.84–1.00) (Appendix A). The cut-off point with the highest predictive value for the development of added PE was 18 mg/dL.

## 4. Discussion

Blood pressure measurement is a marker of quality in prenatal care. The relationship between BP and the development of PE has been widely demonstrated [15], but this relationship is less evident in pregnant women with cHT. The most important finding of our study is that the predictors of PE in the first trimester of pregnancy are different for women with baseline normotension and those with cHT. In pregnant women with cHT, renal laboratory parameters, but not BP values, are the strongest predictors of PE. Secondly, we confirmed that office MBP is the main predictor of PE in normotensive women, with no inferiority with respect to 24-h ambulatory BP parameters.

As early as in the first trimester of pregnancy, we found that office BP is a strong predictor of the development of PE in previously normotensive women, but not in those with cHT. Among all the evaluated hemodynamic parameters, office MBP is the strongest predictor of further PE development. On the contrary, in pregnant women with cHT, neither MBP nor any of the other office or ambulatory BP parameters were good predictors of PE. Similarly, Rovida et al. [12] found a poor predictive value of MBP for sPE in pregnant women with cHT (area under ROC curve, 0.469). Hauspurg et al. [16] described how BP levels at the first visit (11.6 weeks) were independently associated with the risk of PE in a population of 8899 pregnant women at low risk for the development of PE. In this line, Poon et al. [10] demonstrated that increased MBP in the first trimester (11th and 13th weeks of gestation) is closely related to the presence and severity of hypertensive disorder in pregnancy. More recently, Gasse et al. [9] showed that MBP measured in the first trimester is the best predictor of PE. In a meta-analysis [17] that included 34 studies with a total of 60.599 pregnant women, determination of MBP in the second trimester of gestation proved to be the parameter with the strongest predictor of PE in low-risk pregnant women, with an area under the ROC curve of 0.76. We found the same but earlier, in the first trimester, with a higher predictive value. This finding is relevant because it allows for the commencement of prophylactic treatment with acetylsalicylic acid sooner. In our study, normotensive pregnant women who developed PE had a significantly higher BP, both office and 24-h ambulatory, in the first trimester of gestation than those with unaffected pregnancies. This difference was maintained after adjusting for BMI, history of PE in previous pregnancies and Caucasian race. It is possible that this increase in BP in the first trimester mainly reflects the absence of the physiological decrease that occurs at this stage by systemic vasodilation.

However, in these mentioned studies, the prevalence of pregnant women with cHT was almost residual, around 0.5–1.4%. In pregnant women with normal-high BP levels (SBP = 130–139 and/or DBP = 80–89 mmHg) [13] the prevalence of PE was 15.8%, three times higher than in pregnant women with optimal BP (BP < 120 and BP < 80 mmHg) [16]. In our study, in pregnant women with cHT, no significant differences were observed in BP values in the first trimester between pregnant women who developed sPE and those who did not. In contrast, we found that some laboratory parameters, i.e., plasma urea, creatinine, potassium, and uric acid levels determined in the first trimester, were higher in pregnant women with cHT who subsequently developed sPE. Elevation of the concentrations of these laboratory parameters as a consequence of established PE has been widely described in the literature [18] but to our knowledge this is the first time that it is reported to be a predictive marker. In normal pregnancy, the glomerular filtration rate is increased early by 40–60% as a consequence of increased cardiac output and renal plasma flow [19,20]. These observed higher levels in pregnant women who developed sPE as compared to those who did not could result from the absence of increased glomerular filtration. Therefore, based on our findings, early determination of renal parameters, especially urea and creatinine, could help distinguish the pregnant women with cHT with a higher risk of developing sPE than those without.

Of note, we observed a higher percentage of non-Caucasian ethnicity in pregnant women with cHT as compared to normotensive women. Nevertheless, we consider that the results in the renal biochemical parameters are not influenced by this condition, although this should be validated in larger samples.

The second important finding refers to the best method for measuring BP in pregnant women, at least in relation to the prediction of PE. In our study, it is worth highlighting the systematic approach followed to determine office BP. This was carried out strictly in accordance with the recommendations of current HT guidelines [13], i.e., respecting the suggested environmental conditions as well as the rest time and intervals between measurements. This may be important in terms of both this higher predictive ability of MBP in our cohort with respect to other reports and the lack of differences with ABPM that we found. The use of 24 h-ABPM in the general population is considered a very useful tool to predict cardiovascular risk and target organ damage compared to BP readings in the office or self-measurements performed at home [21,22]. However, in current obstetric practice, the performance of 24-h ABPM is not well defined. A recent randomized trial [23] concluded that the use of ABPM in pregnant women reduces induction of labor due to hypertensive causes. In the first trimester, we found that 24-h ambulatory BP parameters have statistically significant differences between pregnant women with PE and those with unaffected pregnancies, but their ability to predict PE was not superior to that of office BP measurements. The main limitation of our study is the small sample size in the cHT group.

In summary, the prediction of PE in the first trimester differs between normotensive pregnant women and pregnant women with cHT. In normotensive pregnant women, the strongest predictor of PE is the determination of BP, and in particular office MBP, with the best cut-off point being 80.3 mmHg. In pregnant women with cHT, biochemical parameters in the first trimester, mostly plasma urea and creatinine, can detect those pregnant women with a higher risk of developing sPE, with the best cut-off values for plasma creatinine and urea being 0.54 mg/dl and 18 mg/dl, respectively. The early detection of the population at higher risk of developing PE or sPE with such easily available markers allows early initiation of prophylactic acetylsalicylic acid treatment before 16 weeks of gestation. Nevertheless, due to defined study limitation, which is number of cases, results should be treated as initial/preliminary findings, whereas so important to be analysed in perspective of bigger study groups.

## 5. Conclusions

Office MBP is the main predictor of PE in normotensive pregnant women, but we point to laboratory parameters as the most important factor to predict sPE in women with cHT. We provide cut-off values for office MBP and plasma urea levels to predict PE in normotensive and cHT pregnant women as determined in the first trimester, respectively. An accurate measurement of office BP is not inferior to 24 h-ABPM to predict PE.

## Figures and Tables

**Table 1 jcm-12-00579-t001:** Baseline characteristics of the study population in the first trimester.

	Normotensive *n =* 124	cHT *n =* 20	*p*-Value
Age (years)	34.2 ± 5.2	34.2 ± 3.8	0.981
Baseline BMI, (kg/m^2^)	24.7 ± 4.2	28.2 ± 5.5	0.001
Primiparous, *n* (%)	56 (45.9)	2 (10)	0.002
Previous PE, *n* (%)	17 (13.9)	5 (26.3)	0.148
Caucasian *n* (%) Non-caucasian, *n* (%)	86 (70) 38 (30)	9 (45) 11 (55)	0.029

Results are expressed as mean ± SD or *n* (%) cHT: chronic hypertension, BMI: body mass index, PE: preeclampsia.

**Table 2 jcm-12-00579-t002:** Comparison of demographic and clinical characteristics between women with or without PE in the first trimester.

	Normotensive			cHT		
	Unaffected (*n =* 116)	PE (*n =* 8)	*p*-Value	Unaffected (*n =* 9)	sPE (*n =* 11)	*p*-Value
Age (years)	34.3 ± 5.1	31.8 ± 5.7	0.199	35.4 ± 2.4	35.4 ± 2.4	0.112
Race, *n* (%)						
Caucasian	83 (71.6)	3 (37.5)	0.026	5 (55.6)	4 (36.4)	0.342
Non-Caucasian	33 (28.4)	5 (62.5)		4 (44.4)	7 (63.6)	
Primiparous, *n* (%)	53 (46.1)	3 (37.5)	0.463	1 (11.1)	1 (9.1)	0.711
Previous PE, *n* (%)	14 (12.3)	3 (37.5)	0.045	0 (0)	5 (50)	0.022
Baseline BMI, (kg/m^2^)	24.51 ± 4.1	28.2 ± 4.1	0.013 0.078	27.5 ± 5.4	28.8 ± 5.9	0.629 0.426
BMI < 30, *n* (%)	102 (87.9)	5 (62.5)		7 (77.8)	7 (63.6)	
BMI ≥ 30, *n* (%)	14 (12.1)	3 (37.5)		2 (22.2)	4 (36.4)	
Family history of hypertension, *n* (%)	52 (45.2)	7 (87.5)	0.023	6 (54.6)	6 (66.7)	0.465
Diabetes, *n* (%)	2 (1.7)	0 (0)	0.888	0 (0)	3 (27.3)	0.145
Dislipidemia, *n* (%)	1 (0.9)	1 (12.5)	0.126	0 (0)	2 (18.2)	0.289

Results are expressed as mean ± SD or *n* (%). BMI: body mass index, cHT: chronic hypertension, PE: preeclampsia, sPE: superimposed preeclampsia.

**Table 3 jcm-12-00579-t003:** Office blood pressure values in the first trimester in normotensive and chronic hypertensive pregnant women.

	Normotensive			cHT		
	Unaffected (*n =* 116)	PE (*n =* 8)	*p*-Value *	Unaffected (*n =* 9)	sPE (*n =* 11)	*p*-Value
Office SBP (mmHg)	104.7 ± 8.9	120.6 ± 10.7	<0.001	130.0 ± 22.1	125.3 ± 11.2	0.552
Office DBP (mmHg)	62.0 ± 7.8	75.7 ± 8.6	<0.001	81.2 ± 14.8	80.9 ± 8.3	0.953
Office MBP (mmHg)	76.2 ± 7.4	90.6 ± 9.1	<0.001	97.4 ± 16.9	95.7 ± 8.8	0.769
Heart rate (bpm)	79.2 ± 10.8	83.3 ± 12.0	0.300	87.4 ± 7.1	84.8 ± 12.0	0.573

Results are expressed as mean ± SD. SBP: systolic blood pressure, DBP: diastolic blood pressure, MBP: mean arterial blood pressure, bpm: beats per minute, cHT: chronic hypertension, PE: preeclampsia, sPE: superimposed preeclampsia. ***** After adjusting for BMI, history of PE in previous pregnancies and Caucasian race.

**Table 4 jcm-12-00579-t004:** Ambulatory blood pressure values in the first trimester in normotensive and chronic hypertensive pregnant women.

	Normotensive			cHT		
	Unaffected (*n* = 116)	PE (*n* = 8)	*p*-Value *	Unaffected (*n* = 9)	sPE (*n* = 11)	*p*-Value
Daytime SBP (mmHg)	111.5 ± 8.7	122.6 ± 7.5	<0.001	131.6 ± 12.8	131.0 ± 12.7	0.909
Daytime DBP (mmHg)	69.0 ± 6.2	77.7 ± 7.0	0.001	80.5 ± 8.1	84.0 ± 7.5	0.341
Daytime MBP (mmHg)	83.0 ± 6.3	92.5 ± 6.6	0.001	97.3 ± 9.7	99.1 ± 8.4	0.656
Daytime HR (bpm)	84.3 ± 8.5	91.2 ± 9.0	0.029	88.0 ± 7.8	82.7 ± 7.3	0.139
Nighttime SBP (mmHg)	99.7 ± 9.4	108.8 ± 5.7	0.008	115.3 ± 17.9	124.8 ± 21.7	0.309
Nighttime DBP (mmHg)	57.8 ± 5.7	64.1 ± 4.4	0.003	67.2 ± 10.1	76.6 ±11.6	0.074
Nighttime MBP (mmHg)	72.1 ± 6.1	80.1 ± 4.2	<0.001	83.6 ± 12.5	92.2 ± 14.5	0.179
Nighttime HR (bpm)	72.5 ± 8.1	81.1 ± 6.8	0.005	74.2 ± 6.1	76.3 ± 6.8	0.476
24-h SBP (mmHg)	107.3 ± 8.3	117.5 ± 6.4	0.001	128.9 ± 15.1	125.2 ± 14.6	0.589
24-h DBP (mmHg)	65.1 ± 5.5	73.1 ± 5.7	<0.001	75.3 ± 8.5	81.7 ± 8.4	0.111
24-h MBP (mmHg)	79.1 ± 5.7	87.8 ± 5.8	<0.001	91.7 ± 10.5	97.0 ± 9.8	0.261
24-h HR (bpm)	80.0 ± 8.0	87.6 ± 7.7	0.011	83.0 ± 6.7	80.6 ± 6.9	0.453

Results are expressed as mean ± SD. SBP: systolic blood pressure, DBP: diastolic blood pressure, MBP: mean arterial blood pressure, HR: heart rate, bpm: beats per minute, cHT: chronic hypertension, PE: preeclampsia, sPE: superimposed preeclampsia. ***** After adjusting for BMI, history of PE in previous pregnancies and Caucasian race.

**Table 5 jcm-12-00579-t005:** Laboratory parameters in the first trimester in normotensive and chronic hypertensive pregnant women.

	Normotensive			cHT		
	Unaffected (*n =* 116)	PE (*n =* 8)	*p*-Value	Unaffected (*n =* 9)	sPE (*n =* 11)	*p*-Value
Glucose (mg/dL)	79.4 ± 11.2	82.8 ± 15.5	0.414	76.7 ± 6.1	101.5 ± 50.1	0.478
Urea (mg/dL)	17.4 ± 4.6	18.2 ± 4.4	0.643	15.0 ± 2.6	21.1 ± 3.6	0.001
Creatinine (mg/dL)	0.5 ± 0.1	0.4 ± 0.1	0.906	0.4 ± 0.1	0.6 ± 0.1	0.004
Urate (mg/dL)	2.8 ± 0.5	2.6 ± 0.4	0.391	3.1 ± 0.5	4.3 ± 1.4	0.041
Sodium (mmol/L)	137.1 ± 1.9	137.0 ± 1.6	0.839	137.1 ± 2.4	136.3 ± 2.0	0.464
Potassium (mmol/L)	4.0 ± 0.2	4.0 ± 0.2	0.733	4.0 ± 0.2	4.2 ± 0.2	0.044
Total Cholesterol (mg/dL)	180.6 ± 26.6	177.4 ±36.7	0.761	185.7 ± 33.6	169.5 ± 30.5	0.332
Triglycerides (mg/dL)	105.0 ± 48.7	106.1± 63.7	0.953	130.7 ± 68.6	185.1 ± 94.5	0.222
hs-CRP (mg/dL) *	0.57 [0.02–4.8]	1.06 [0.1–3.3]	0.260	1.0 [0.1–1.9]	2.1 [0.38–2.9]	0.222
UACR (mg/g) *	3.9 [1.2–27]	3.6 [2–5–3.2]	0.565	4.3 [2.8–18]	11 [2.9–145]	0.173

* Data expressed as median [IQR]. All other parameters are expressed as mean ± SD. hs-CRP: high-sensitivity C-reactive protein, UACR: urine albumin/creatinine ratio, cHT: chronic hypertension, PE: preeclampsia, sPE: superimposed preeclampsia.

**Table 6 jcm-12-00579-t006:** Area under ROC curve (95%CI) of BP parameters for the subsequent development of PE in normotensive and hypertensive pregnant women.

	Normotensive		cHT	
	Area under ROC Curve	95%CI	Area under ROC Curve	95%CI
Office SBP (mmHg)	0.89	(0.80–0.98)	0.50	(0.19–0.80)
Office DBP (mmHg)	0.89	(0.79–0.98)	0.50	(0.21–0.79)
Office MBP (mmHg)	0.91	(0.82–0.99)	0.47	(0.16–0.78)
Daytime SBP (mmHg)	0.83	(0.72–0.95)	0.46	(0.19–0.74)
Daytime DBP (mmHg)	0.82	(0.70–0.95)	0.65	(0.39–0.91)
Daytime MBP (mmHg)	0.84	(0.73–0.95)	0.59	(0.32–0.87)
Nighttime SBP (mmHg)	0.83	(0.75–0.91)	0.68	(0.41–0.95)
Nighttime DBP (mmHg)	0.81	(0.70–0.92)	0.74	(0.49–1.00)
Nighttime MBP (mmHg)	0.85	(0.78–0.94)	0.69	(0.42–0.95)
24-h SBP (mmHg)	0.84	(0.74–0.94)	0.61	(0.33–0.88)
24-h DBP (mmHg)	0.85	(0.75–0.95)	0.73	(0.48–0.97)
24-h MBP (mmHg)	0.86	(0.77–0.96)	0.64	(0.37–0.92)

SBP: systolic blood pressure, DBP: diastolic blood pressure, MBP: mean arterial blood pressure, cHT: chronic hypertension, CI: confidence interval.

**Table 7 jcm-12-00579-t007:** Area under curve ROC (95%CI) for the prediction of PE by determination of office MBP in the first trimester in normotensive pregnant women.

	Area under ROC Curve	95%CI	Sensitivity (%)	Specificity (%)	PPV (%)	NPV (%)	Best Cut-Off
Office MBP (mmHg)	0.91	(0.82–0.99)	100	74	21	100	80.3

MBP: mean arterial blood pressure, CI: confidence interval, PPV: positive predictive value, NPV: negative predictive value.

**Table 8 jcm-12-00579-t008:** Area under curve ROC for the prediction of PE according to laboratory parameters in the first trimester in pregnant women with chronic hypertension.

	Area under ROC Curve	95%CI	Sensitivity (%)	Specificity (%)	PPV (%)	NPV (%)	Best Cut-Off
Urea (mg/dL)	0.94	(0.84–1.00)	90.9	87.5	90.9	87.5	18
Creatinine (mg/dL)	0.93	(0.83–1.00)	81.8	100	100	80	0.54

CI: confidence interval, PPV: positive predictive value, NPV: negative predictive value.

## Data Availability

Data available upon request from authors.

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
