# Peer review of "Predictors of Preeclampsia in the First Trimester in Normotensive and Chronic Hypertensive Pregnant Women"

_jcm, 2023, doi:10.3390/jcm12020579_

Round 1

Reviewer 1 Report

In this article, Vasquez et al. considered several hemodynamic and biochemical parameters measured in women during their first trimester of pregnancy so as to determine those which are the most predictive of preeclampsia onset. Notably, noting that known predictors perform poorly if women suffer from chronic hypertension, the authors aimed to investigate whether the best predictive parameters were different if women were normotensive during early pregnancy or if they suffered from chronic hypertension. The study was carried out on an unbalanced design (124 normotensive women, 20 women with chronic hypertension).

Given the increase incidence of chronic hypertension among pregnant women, the objectives of the study are of interest to help monitor these women during their pregnancy and anticipate preeclampsia onset. Nonetheless, I think several points of the manuscript need be revised.

Here are my comments and suggestions on specific points of the manuscript:

-       Abstract: The abstract is quite difficult to read and I think it would gain in being re-written so as to focus on the main points of the study. I would expect it to follow the usual framework: background, objectives of the study methods and results. Objectives and methods are not clearly presented in the current abstract and are only discussed at the end of the introduction (lines 53-56). Moreover, the abstract contains many figures and data statistics which should rather be described in the result section, not in the abstract (lines 16 to 22 should rather be replaced by a short description of the method). Also: the acronym “chronic-HTN” should be clarified.

-       Line 42: Reference [6] needs be reformatted.

-       Line 46 and several times in the manuscript: MBP was suggested to be a good predictor of preeclampsia, not necessarily “the best” one. Pulse pressure (SBP-DBP) was also suggested as an early predictor of preeclampsia (see for example: Pulse Pressure and Risk of Preeclampsia by Thadani et al.). This predictor is independent of MBP. Why was this indicator not investigated in the study? Could you consider including it along with MBP in this study?

-       Results section: lines 101 to 107 are duplicated in lines 108 to 114.

-       Line 122: the rate of PE is high among chronic-HTN women. If I understand well the methods, it is mainly (maybe only) proteinuria (urine protein/creatinine ratio >0.3) which is used to “diagnose” PE in these women. It is thus obvious that this renal parameter will perform better than BP to predict preeclampsia in chronic-HTN women. To me, the fact that a potential predictor of a disease is specifically used to diagnose the disease will surely bias the analysis and this is a major flaw in the study which should at least be mentioned in the Discussion section. Nonetheless, this bias could be mitigated if a diagnostic of PE made by a clinician during pregnancy was available and used to define PE. Do you have this information?

-       The very low number of chronic-HTN women (20) in a strong limitation of this study which should appear in the Discussion section.

Reviewer 2 Report

The idea of the study is interesting as PE is a serious clinical problem in which diagnostic possibilities are still insufficient. Topic should be deeply studied and further experiments are really needed.

I have one substantial problem with revised paper. The results are not that convincing for me if we realize that your study group was actually not 144 patients with more/less equal subgroups, but "116" vs. "28" or precisely 9 vs. 11 ... It may be finally treated as preliminary study to show the trend and speculate if it will be visible and statistically significant on bigger group... Additionally, you analyze many data, that's nice, but in such asymmetric groups why you decided to compare mean values not medians?

Other small remarks. I would expect only up-to-date papers in the references as this is really hot topic and there is plenty studies for citations. Moreover, in my perspective presentation of figure 1 and 2 is useless.

Personally, I don't like abbrev. HTN. Maybe HT. And why not use cHT for chronic hypertensive?

Round 2

Reviewer 2 Report

Thank you for your responses.

I partially agree with them. I mean, verification of normality refers to usage of mean value and reflects incidence in the population, that's all good, however it doesn't correspond with correlation tests usage in such huge assymetry.

I would implement phrase that due to defined study limitation, which is number of cases, results should be treated as initial/preliminary findings, whereas so important to be analyzed in perspective of bigger study groups.

I would recommend paper for publication.